# Enhancement of Epoxy Thermosets with Hyperbranched and Multiarm Star Polymers: A Review

**DOI:** 10.3390/polym14112228

**Published:** 2022-05-30

**Authors:** David Santiago, Àngels Serra

**Affiliations:** 1Eurecat–Chemical Technologies Unit, C/Marcel·lí Domingo 2, 43007 Tarragona, Spain; 2Department of Mechanical Engineering, Universitat Rovira i Virgili, Av. Països Catalans 26, 43007 Tarragona, Spain; 3Department of Analytical and Organic Chemistry, Universitat Rovira i Virgili, C/Marcel·lí Domingo 1, 43007 Tarragona, Spain; angels.serra@urv.cat

**Keywords:** hyperbranched polymers, multiarm star polymers, epoxy resins, thermosets

## Abstract

Hyperbranched polymers and multiarm star polymers are a type of dendritic polymers which have attracted substantial interest during the last 30 years because of their unique properties. They can be used to modify epoxy thermosets to increase their toughness and flexibility but without adversely affecting other properties such as reactivity or thermal properties. In addition, the final properties of materials can be tailored by modifying the structure, molecular weight, or type of functional end-groups of the hyperbranched and multiarm star polymers. In this review, we focus on the modification of epoxy-based thermosets with hyperbranched and multiarm star polymers in terms of the effect on the curing process of epoxy formulations, thermal, mechanical, and rheological properties, and their advantages in fire retardancy on the final thermosets.

## 1. Introduction

Dendritic polymers have attracted great interest during the last 40 years due to their unique characteristics, facile structural modification, and potential applications. Among them, hyperbranched polymers (HBPs) have been the most exploited in technological applications at large scales, but multiarm star polymers (MASPs) have also played an important role. Since HBPs were first studied in the 1980s with the publication of some patents and research papers [1,2,3], the number of publications regarding the utilization of these materials has continually increased to 2010 (Figure 1). Although the emergence of new technologies such as nanomaterials or green technologies has attenuated the interest in HBPs, they are still in the crosshairs of academics and industries.

HBPs are a subclass of dendritic polymers that can be effectively used as modifiers of thermosetting polymers because of their interesting properties resulting from the branched architecture and a high number of functional groups [4]. They have a highly randomly branched structure which makes them less viscous than their linear counterparts of the same molecular weight because of the absence of chain entanglements [5,6]. HBPs have a high number of reactive end groups that can be modified to enhance the physical compatibility between the HBP and the matrix or to become covalently attached to the network. The properties of the final materials can be tailored as a function of HBPs’ chemical structure, average molecular weight, degree of branching, or type of final functional groups. The tailored structure of HBPs allows the enhancement of the mechanical dielectric, and optical properties of the final materials, their thermal degradability, or fire retardancy among others [7].

Dendrimers are the most representative type of dendritic polymers, and their perfect structure defines the group’s architecture. They have a complete regularly branched and uniform structure (Figure 1a), but their synthesis involves multiple reaction steps and complicated synthetic processes and purifications, which make them very expensive [8,9]. On the contrary, HBPs (Figure 1b) can be easily synthesized through a one-step polymerization reaction and possess similar properties to dendrimers in general technological applications. However, their simple synthetic process may limit the control of molar mass and branching accuracy and leads to heterogeneous products with a wide molar mass and branching distribution [4,10].

Perfectly branched dendrimers are suitable for applications in medicine as drug carrier molecules or in gene delivery, as standards or models for biomolecules, or as catalytic active molecules [7]. However, in the case of HBPs, the lack of a well-defined structure and molar mass is a disadvantage in these sensitive areas. In addition, their nature limits their application as bulk materials [11]. Nevertheless, HBPs have attracted substantial interest as a blend component, melt modifier, or additive due to their characteristics in numerous fields of applications, such as coatings, multifunctional crosslinkers, or surface modifiers [12,13,14].

Based on a hyperbranched core structure, MASPs have attracted the attention of many researchers [15], some of them adopting these structures as additives for epoxy materials allowing improvements in their properties [16]. Their structure is depicted in Figure 1c. Their synthesis is quite straightforward from a hyperbranched core by growing linear polymers of different chemical nature using the chain end groups of the HBP as initiating polymerization points. This is called core-first methodology. Alternatively, starting from the linear polymers, which constitute the arms, by a condensation or coupling reaction, they can be linked covalently to the end-functionalized hyperbranched structure that acts as a core. This synthetic methodology is known as arm-first. MASPs, although not as common as HBPs, have also been used as additives for epoxy thermosets. Their complex structure facilitates the phase separation in the epoxy matrix which helps to improve the toughness or flexibility of the network structure [16,17].

In this review, the influence of HBPs and MASPs on the physical properties of epoxy thermosets is summarized through well-known studies on the topic, as well as through some of the most recent articles. Numerous reviews have been published regarding different aspects of HBPs and MASPs, such as synthesis, modification, and applications [18,19,20,21,22,23,24,25]. However, only a few articles review the modification of epoxy thermosets with HBPs [26,27] and none with MASPs as additives.

## 2. Epoxy Thermosets

Epoxy resins are widely used in applications such as coatings, adhesives, structural applications, or electronics because of their good mechanical properties, relatively low shrinkage, and high chemical and thermal resistance [28]. The wide range of curing agents available for their crosslinking makes them extremely versatile. The higher crosslinking density of epoxy thermosets allows them to reach high glass transition temperatures (*T_g_*s) and high hardness, tensile strength, shear strength, and Young’s modulus [29]. However, their inherent brittle behavior limits their range of applications, especially for structural purposes. Their high crosslinking density reduces the impact strength and lowers their resistance to crack propagation [30]. During the curing reaction, the formation of a great number of covalent bonds produces a shrinkage, which finally leads to the apparition of internal stresses that originates cracks, voids, warping, and delamination in composites. The scratch resistance is another concern to be considered [29]. Strategies to overcome all these drawbacks without compromising the advantages of epoxy thermosetting materials have been explored for decades.

There are many methodologies to enhance the toughness of epoxy resins. The more extensively used is the addition to the formulation of thermoplastics, block-copolymers, liquid rubbers, inorganic particles, core-shell particles, or nanoparticles [31,32,33,34,35,36,37,38,39,40,41,42]. However, all these additives behave with some drawbacks that must be considered. The initially miscible mixture of epoxy/hardener becomes immiscible with the added modifier as the curing reaction proceeds due to a growing incompatibility between the developing network and the modifier. It is well known that the modification of an epoxy system must not lead to phase segregation or filtering out modifiers induced by phase instabilities [26]. Thus, good compatibility between the matrix and the polymeric modifier is necessary for effective toughening [43]. Butadiene/acrylonitrile copolymers are successful tougheners of epoxy resins but they have a high level of unsaturation in their structure which provides sites for degradation reactions in oxidative and high-temperature environments [44]. Liquid rubbers have also been used as epoxy modifiers but their use often results in a viscosity increase and a notable decrease in the final *T_g_* [44]. Moreover, the modification of epoxy thermosets with inorganic particles often requires the use of coupling agents to improve the compatibility of inorganic particles and matrices [45].

HBPs and MASP can be used as effective polymer modifiers of thermosetting epoxy-based materials because of the advantageous features discussed above. The improvement in the toughness is attributed to the flexibilizing effect induced by the homogeneous incorporation of the HBP or to the local inhomogeneities created in the crosslinked network by the formation of a phase-separated morphology, which hinders the propagation of cracks [29]. In addition, the properties of the final material can be tailored as a function of the structure, average molecular weight, degree of branching, and the type of functional end-groups to obtain the desired properties. In the following sections, the effect of HBPs and MASPs on some aspects of epoxy thermosets are discussed.

## 3. Effect of the Addition of HBPs or MASPs on the Curing Process

The curing kinetics of hyperbranched-modified epoxy-based thermosetting materials is complex and depends on the curing agent/initiator used, the type and number of reactive functional groups of the HBP, its molecular weight, and the changes in viscosity and mobility modification produced by the characteristics of the HBP added. There is no consensus on how the addition of HBPs affects the curing kinetics. Although the presence of polymeric modifiers such as HBPs increases the viscosity of the formulations, it is not foreseeable if the reaction rate will decrease, by the reduced mobility or the topological restrictions, or it will decrease because of the effect of the HBPs end groups on the curing mechanism [43].

As mentioned before, epoxy resins are extremely versatile due to the wide range of available curing agents. One of the most studied systems is the epoxy/amine, which in some previous works was modified by adding hydroxyl-ended HBP polyesters [46,47,48]. In these systems, HBPs were used as an additive with contents up to 40%. The blends were usually prepared by blending the HBP and the epoxy resin and then adding the amine curing agent to obtain the stoichiometric ratio of 1:1 between N-H and epoxy groups. In these systems, HBPs do not react with epoxy resins and act as an additive which could eventually enhance toughness. The addition of HBPs apparently did not change the curing reaction mechanism since the enthalpy of the reaction decreased proportionally to the HBP content. Rozenberg [49] reported for the first time the catalytic effect of OH groups in epoxy/amine systems, and therefore, it is foreseeable that the addition of hydroxylated HBPs can enhance the curing rate.

Choosing suitable end groups or curing agents can allow the enhancement of the compatibility between HBPs and epoxy resins by the formation of covalent bonds. For instance, epoxy groups are reactive in amine curing systems while hydroxyl groups are not, at least at conventional curing temperatures (<250 °C). However, when hydroxyl-ended HBPs are used under cationic conditions, they become incorporated into the network structure through hydroxyl-induced chain transfer reactions that occur, via monomer-activated mechanism, in the cationic ring-opening polymerization of epoxy groups [50,51,52].

The epoxy/anhydride system modified with hydroxyl-ended HBPs is another interesting system that has been widely studied [53,54]. The anhydride enhances the miscibility of hydroxyl-ended HBPs in the epoxy matrix, the primary alcohols of the HBP increase the epoxy/anhydride curing rates, and the anhydride group would react with the hydroxyl groups of HBP leading to more homogeneous thermosets [53]. The addition of hydroxyl-ended HBPs provides ester linkages by reaction of anhydride with epoxides and with hydroxyl groups, which enhance the properties of the materials and barely affect curing kinetics. Because the epoxy/anhydride reaction is slow, an accelerator, such as a tertiary amine, was often used to initiate the process at lower temperatures. In the case of modification of epoxy/anhydride systems with hydroxyl-terminated HBPs, the reactive process is complex and involves several reactions: esterification of the hydroxyl groups in the chains of the HBP and those formed from the attack of the carboxylic groups previously formed to the epoxy groups, the alternate polymerization of anhydrides and epoxides initiated by the tertiary amines, and the polycondensation of hydroxyl and carboxylic groups.

Foix et al. [53] studied the effect of different proportions of commercial hydroxyl-terminated HBP polyester Boltorn^®^ H30 (Figure 2) (Perstorp, Malmö, Sweden) on the curing process of epoxy/anhydride systems catalyzed with a tertiary amine. The addition of HBP clearly changed the shape of the curing exotherm in the non-isothermal calorimetry experiments (Mettler Toledo DSC 821e, Columbus, OH, USA), shifting the peaks to lower temperatures. However, DSC thermograms did not show many differences in shapes and temperatures when changing the proportions of HBP. The study of the conversion rate with temperature in the presence of hydroxylic-HBP revealed an acceleration of the curing process in the early stages in comparison with the neat epoxy/anhydride system. However, from the 80% of conversion, the HBP showed a retardant effect, and the complete curing was only achieved at high temperatures. Similar results were obtained with the commercially available hyperbranched poly(ester amide) Hybrane^®^ S1200 and S2200 (Figure 3) (Covestro, Leverkusen, Germany) [55,56]. In addition, increasing the proportion of ester groups in the thermosets improved their thermal reworkability by saponification.

The chemical incorporation of hydroxyl-terminated HBPs into the epoxy matrix can also be accomplished through anionic ring-opening polymerization. Morell et al. [57] synthesized a hyperbranched polyaminoester and used it to modify DGEBA resin with 1-methylimidazole as anionic initiator. Hydroxyl groups facilitated the attack of the tertiary amine on the oxirane group, enhancing the initiation rate. An acid-base equilibrium was also established between the hydroxyl groups of the HBP and the alkoxide formed by the attack of the tertiary amine on the epoxide. This resulted in the formation of alkoxides as terminal groups in the HBP and allowed them to actively participate as chain transfer agents promoting the covalent linkage of the HBP to the epoxy matrix. The curing process barely changed when increasing the HBP content. The presence of hyperbranched polyaminoester slightly reduced the reactivity of the system at low conversions but increased it from a certain value.

Hydroxyl-ended HBPs represent most of the research carried out on the toughening of epoxy resins with HBPs. Although there are a lot of interesting reports focused on the synthesis of amino-terminated HBPs [58,59,60,61,62,63], less work has been conducted on the effect on the curing kinetics in epoxy systems. Jin and Park [64] synthesized an amine-ended HBP from the condensation polymerization of 2,4,6-triaminopyrimidine with 4,4-biphthalic anhydride and used it to modify a DGEBA resin. The polymerization reaction was catalyzed by 4,4′-diaminodiphenyl methane. The exothermic peak in the DSC thermograms shifted to lower temperatures with the addition of this HBP and the 4 wt.% HBP modified DGEBA showed a higher conversion than neat DGEBA at the same curing times. These results were discussed in terms of the addition of highly reactive amine groups of the HBP to the epoxy matrix. The reactivity of the amine groups with epoxides resulted in an increase in the reaction rate in the DGEBA/HBP blends in comparison with the neat DGEBA formulation. Similar results were obtained in the study by Fernández-Francos et al. [65]. The authors used commercially available hyperbranched poly(ethyleneimine)s Lupasol^®^ (BASF, Ludwigshafen am Rhein, Germany) (Figure 4) as polymeric modifiers in DGEBA thermosetting formulations using 1-methylimidazole (1MI) as anionic initiator. The amine functional groups of the HBP together with tertiary amines of 1MI in epoxy formulations had a synergistic effect in terms of reaction kinetics because of the generation of hydroxyl groups by the epoxy/amine condensation.

Epoxy-ended HBPs are another important class of tougheners for epoxy thermosets and have shown excellent results regarding the toughening effect and mechanical properties [66,67,68,69], which are discussed elsewhere. However, the study on the curing kinetics of epoxy systems modified with epoxy-ended HBPs has generated less interest. Some research studies used the commercially available epoxy-ended aliphatic-polyester HBP Boltorn^®^ E1 and E2 (Perstorp, Malmö, Sweeden) (Figure 5). In the studies by Mezzenga et al. [27] and Ratna et al. [44], the authors used Boltorn^®^ E1 as a modifier of a DGEBA thermoset cured with diamines. The HBP was compatible with DGEBA at the curing temperature used (100 °C), but as the reaction progressed, the epoxy groups of DGEBA reacted faster than the epoxy groups of the HBP. The curing reaction produced hydroxyl groups which decreased the compatibility of DGEBA and HBP leading to the formation of two-phase microstructures. The HBP did not modify the curing mechanism as the total enthalpy was close to the expected theoretical value [44].

There is evidence that a reaction occurs between epoxy groups of the epoxy-ended HBP with the epoxy network in epoxy/anhydride systems [27,70,71]. This reaction takes place at higher temperatures than the main epoxy curing peak, which suggests that the reactivity is lower than that of the developing network. This difference in reactivity has been attributed to the lack of mobility of the epoxy groups in the HBP caused by topological constraints and the inability of the hydroxyl groups in the HBP to catalyze the curing reaction. Notwithstanding, the reaction between epoxy-ended HBPs and epoxy resins suggests that materials must have good compatibility, which is an important factor for the toughening of epoxy resins.

UV-polymerization has attracted substantial interest due to its wide fields of applications such as printing inks, adhesives, or composites [72,73,74]. This technology uses light irradiation to initiate photochemical and chemical reactions, leading to the formation of new polymeric materials. UV-polymerization possesses many advantages up front compared to traditional thermal curing: faster polymerization rates, lower energy consumption because it is usually carried out at room temperature, and it is environmentally friendly due to the use of solvent-free formulations [75]. Sangermano’s group studied the use of different HBPs in the cationic photopolymerization of epoxy resins. In the research carried out with a cycloaliphatic epoxy resin modified with a hydroxyl-ended phenolic HBP [76] and with Boltorn^®^ H20, H30, and H40 [77], a notable increase in curing rates and in the final epoxy groups conversion were obtained in formulations modified with HBPs with respect to the neat epoxy. These results were attributed to the chain-transfer reaction involving hydroxyl groups present in the structure of the HBPs. This mechanism resulted in polymer chain-length and crosslinking density decrease. As a result, network structures became more flexible, and the mobility of the reactive species increased. That led to a delay in the vitrification and a higher epoxy group conversion. Opposite results were obtained when the cycloaliphatic epoxy resin was modified with MASPs based on hyperbranched polyester core and poly(ester methacrylate) arms [16]. A decrease in the photopolymerization rate and epoxy group conversion was observed with increasing HBP content. This result was attributed to an increase in viscosity and a faster vitrification effect. Morancho et al. [78,79] compared the curing kinetics of a DGEBA resin modified with the commercial Boltorn^®^ H20 and Boltorn^®^ H40 as toughening agents using thermal curing and UV curing. Low contents of HBP decelerated the reaction but an acceleration was observed with 20 wt.% when thermal curing was selected. As mentioned before, the opposite effects of increased viscosities and increased number of hydroxyl groups explained these results. When UV curing was used, the temperature played a very important role. At low temperatures, the solubility of the HBP was poor and the curing reaction was hindered. However, at higher temperatures, the HBP was completely solubilized and promoted the curing reaction.

More recently, Xia et al. [80] studied the toughening effect of the hydroxyl-ended Boltorn^®^ H2004 and a poly(tetramethylene ether glycol) (both with molecular weight around 2000 g/mol) on a cycloaliphatic epoxy resin cured with UV-light. The photopolymerization kinetics were significantly faster with the HBP than with the linear polyether. Reactive OH groups of both modifiers had a positive effect on the cationic UV-curing of the epoxy, but the HBP had a larger amount of hydroxy groups exposed to the surface layer while the random coil morphology of the linear poly(tetramethylene ether glycol) limited the number of active OH groups topologically available.

Our research group investigated the effect of allyl-terminated HBPs as modifiers in epoxy thermosets. The first studies were carried out by partially esterifying the OH groups of commercial Boltorn^®^ H30 with 10-undecenoyl moieties (Figure 6) [81,82]. These vinyl-modified HBPs were used as additives in DGEBA formulations cured with Yb(OTf)_3_ as cationic initiator and in epoxy/anhydride systems. The effect of the HBPs on the curing kinetics depended on the amount of reactive OH groups and the changes in viscosity and mobility of the structure that were originated from different degrees of modification. The DSC thermograms revealed no significant changes in comparison with neat epoxy formulations because of the opposite influence of the acceleration effect of hydroxyl-ended HBP in cationic epoxy polymerization and anhydride systems and the reduced mobility of the network due to the increase in viscosity.

Considering all the experience with hyperbranched polyesters and their modification to introduce allyl functional groups, our research team managed to synthesize modified hyperbranched poly(glycidol)s to combine all the good characteristics previously obtained with the modification of Boltorn^®^ H30 with the flexible and polar structure of this HBP. Flores et al. [83] used 10-undecenoyl derivatized hyperbranched poly(glycidol)s to modify a cycloaliphatic epoxy resin, and Yb(OTf)_3_ was selected as thermal cationic initiator. The addition of the HBP with an 80% degree of modification produced a delay in the curing process and the effect was more important when the degree of modification was 45%. The positive contribution by the hydroxyl groups of the HBP was offset by the formation of less active cations, favored by the abundance of ether linkages in the HBP structure, lowering the curing process. The HBP with a 45% degree of modification formed a higher amount of ether groups which explained the higher delay in comparison with the HBP with 80% of modification.

Tomuta et al. [84] synthesized novel HBPs formed by an aliphatic-aromatic hyperbranched polyester core and a shell with aliphatic chains of 10-undecenoyl moieties and allyl groups (Figure 7) and they were used as a modifier of DGEBA/anhydride formulations with a tertiary amine as a catalyst. An acceleration with all modifiers could be observed due to the existence of residual carboxylic acid and the presence of tertiary amines. HBP-allyl accelerated to a higher extent than HBP-undecenoyl because of the different viscosities of the mixtures and the different proportions of residual carboxylic acid of the focal point of the HBP. These materials were homogeneous and showed a modest increase in impact strength. In addition, the HBPs led to a considerable decrease in the *T_g_*. To solve these issues, the same authors used allyl and 10-undecenoyl modified hyperbranched poly(glycidol) as modifiers of DGEBA thermosets cured by adipic dihydrazide [85]. Dihydrazides possess a high crystalline character and they are difficult to disperse or dissolve because melting the curing process is extremely fast. The addition of the allyl and 10-undecenoyl-ended HBPs to the epoxy formulation increased the toughness of the thermoset and helped to disperse adipic dihydrazide in the DGEBA resin. In addition, the HBP led to a significant curing rate reduction, and thus higher curing temperatures were needed to reach complete curing.

The modification of hydroxyl-ended hyperbranched polymers led to a significant enhancement of mechanical and thermomechanical properties without significant changes in the curing kinetics. 

Our research group also reported the modification of the commercial hyperbranched poly(ethyleneimine) with amino groups Lupasol^®^ by reacting with allyl glycidyl ether (Figure 8) or with 10-undecenoyl chloride [86,87,88]. The hyperbranched Lupasol^®^ was partially derivatized with 10-undecenoyl groups and was used as a modifier of DGEBA formulations cured with anhydride in the presence of a tertiary amine as a catalyst. The DSC thermograms did not show significant changes when the modified HBP was added. The calculated activation energy decreased proportionally to the amount of HBP in the formulation. The lowest value was obtained in the formulation with 20% of HBP with a degree of amine conversion of 78%. The acceleration of the curing kinetics was caused by the remaining unreacted amine groups in the HBP, which could react with epoxy or anhydride groups, leading to hydroxyl or carboxyl chain ends that can participate actively in the curing process and even have a catalytic effect [86].

Several studies about the synthesis or modification of HBPs can be found in the literature to introduce other reactive groups in their structure and their impact on curing kinetics. Santiago et al. [43] studied the addition of hyperbranched poly(glycidol) (PG) and its benzoylated derivative (PGBz) (Figure 9) as a modifier of a DGEBA cured with a cationic initiator. Formulations with PGBz produced a gradual decrease in the curing rate but this effect was significantly lower than in the case of formulations with PG. This decrease in the curing rate was caused by the increased viscosity of the formulations with hyperbranched poly(glycidol)s. However, the important difference shown between formulations with PG and PGBz could be related to the difference in compatibility because of the different concentrations of hydroxyl groups, which turns the hydrophilic character of PG to hydrophobic of PGBz.

Morancho et al. [89] synthesized novel poly(ethyleneimine)s with t-butyl and phenyl as terminal groups (Figure 10) and used them as a modifier in DGEBA/anhydride systems with a tertiary amine as initiator. The HBPs were obtained by the modification of commercial Lupasol^®^ with phenyl isocyanate (PEI-PhNCO) to block all the N-H groups and two different proportions of t-butyl isocyanate to obtain a degree of modification of 75% and 100% (PEI-BuNCO75 and PEI-BuNCO100). The kinetic study revealed no changes in the curing reaction of DGEBA/hexahydro-4-methylphythalic anhydride formulations modified with PEI-BuNCO100, an acceleration effect with PEI-BuNCO75 and a deceleration with PEI-PhNCO. The acceleration effect, when PEI-BuNCO75 was used, was attributed to the unreacted N-H groups in the modifier structure, which could react with epoxides and anhydrides because of their nucleophilic character. The deceleration observed was explained based on the dilution effect of reactive groups and the increase in viscosity on adding PEI-PhNCO.

In the study carried out by Fei et al. [90], the authors synthesized a carboxyl-modified HBP (Figure 11) to be used as a modifier in an epoxy/anhydride system. The incorporation of HBP-COOH into the epoxy resin revealed a slight acceleration of the curing reaction of the epoxy/anhydride system. The HBP-COOH contained abundant carboxyl and hydroxyl groups generated by ring-opening polymerization of epoxy groups. The terminal carboxyl groups could act as hardeners and react with the epoxy groups, and the hydroxyl groups could produce an acceleration of the curing process for epoxy/anhydride systems by initiating the polycondensation mechanism.

HBPs can also be modified to act as multifunctional crosslinker agents, having the role of macroinitiator and toughening agent at the same time. HBPs, prepared by thiol-ene click chemistry, could be used as a thermal latent multifunctional macroinitiator. Click chemistry has been used for the last few decades as a polymerization system to form new thermosetting materials with enhanced characteristics. Click-type reactions are fast, can be performed in an air atmosphere, can be triggered either thermally or photochemically, and have high yields without secondary reactions [91]. In the studies carried out by Foix et al. [92] and Flores et al. [93], HBPs were used as modifiers in thiol-ene processes to form a macroinitiator with multiple initiating sites for the homopolymerization of an epoxy resin. First, a reaction took place at room temperature by photoirradiation of the formulation which generated new sulfide bonds between thiols and the epoxy groups. The sulfide groups reacted with the oxonium terminated growing chains to form trialkyl sulfonium salts. The activation of these epoxy groups occurred with UV light by the presence of a cationic photoinitiator. These sulfonium salts were inactive under UV light but could initiate the thermal curing of cycloaliphatic epoxy resins. Acebo et al. [94] synthesized a multifunctional triazole initiator from the hyperbranched poly(ethyleneimine) Lupasol^®^ (Figure 12) by the azide-ine click reaction. The calorimetric studies revealed that the curing of DGEBA formulations with the HBP showed lower reactivity in comparison with 1-methylimidazole.

To analyze the influence of the degree of branching on the curing kinetics of epoxy thermosets, Foix et al. [95] studied three HBPs with different degrees of branching on the cationic curing of DGEBA resins with Yb(OTf)_3_ as cationic initiator. At low heating rates, two peaks in the calorimetric curves were observed. The peak at lower temperatures increased upon increasing the degree of branching. Since the percentage of hydroxyl groups in the formulation increased with the degree of branching, this peak could be associated with the propagation taking place by the activated monomer mechanism. The curing process was slightly decelerated on increasing the proportion of HBP or decreasing the degree of branching. The values of enthalpy released per epoxy equivalent confirmed that the degree of curing achieved was similar for all the formulations studied and was almost complete.

The effect of HBPs on the curing kinetics of different epoxy systems then becomes clear. However, it is interesting to compare the effect between HBPs and their linear counterparts. In the studies carried out by Santiago et al. [96] and Fernández-Francos et al. [65], the effect of the commercial hyperbranched poly(ethyleneimine) Lupasol^®^ on the curing kinetics of a DGEBA and the properties of the final thermosets in comparison with the use of an analogous amine (diethylenetriamine, DETA) was put into evidence. The curing mechanism was similar to both amines and followed a general epoxy/amine polycondensation mechanism. However, the curing rate was slower using the HBP polymer than the linear DETA because of the lower mobility of HBP. Gelation took place earlier with HBPs because of the higher number of functional groups, which reduce the conversion at the gel point.

As previously stated, MASPs are another subclass of dendritic polymers that consist of linear polymeric chains radiating from one single branched point or core [97]. The low cost of HBPs, the high number of reactive functional groups, and their easy synthesis make them adequate cores for the preparation of MASPs. MASPs with hyperbranched polyester and hyperbranched polystyrene core with poly(ε-caprolactone) arms have been reported as modifiers in DGEBA formulations [98,99,100,101] (Figure 13). There was not much influence on the temperature or shape of the curing exotherm by changing neither the proportion of modifier nor the arm lengths, but a slight deceleration was observed in the conversion obtained from DSC and FTIR analysis on increasing the proportion of modifier. The retarding effect was more pronounced when the arms of the MASP were longer, and this was attributed to the effect of the increased viscosity of the mixture. It was expected that the addition of hydroxyl-ended star polymers to the formulation increased the curing rate, but the addition of these modifiers decreased the concentration of epoxy groups and produced a deceleration instead. Another factor that could contribute to this behavior was that the hydroxyl groups of the MASPs could be associated internally with the carbonyl groups of the poly(ε-caprolactone) arms reducing their activity in this way [101].

Acebo et al. [102] modified a hyperbranched poly(ethyleneimine) by incorporating poly(ε-caprolactone) arms (Figure 14). According to DSC studies, there was not much influence on curing kinetics. A more detailed inspection of the thermograms revealed a slight acceleration at the beginning of the curing reaction and a deceleration at the end. This effect was more pronounced on increasing the length of the arms of poly(ε-caprolactone). This behavior could be attributed to the existence of a higher content of hydroxyl groups in modified formulations in comparison with the neat (which increased the curing rate at the beginning of the curing reaction). Moreover, the addition of this modifier increased the viscosity of the formulation and decreased the concentration of reactive species both leading to a retarding effect. Again, the values of enthalpy per epoxy equivalent indicated an almost complete polymerization.

MASPs with poly(styrene) and poly(methyl methacrylate) arms were also used as modifiers in epoxy systems [103,104] (Figure 15). The addition of small quantities of these modifiers (~5 wt.%) did not severely affect the curing process since no chemical incorporation of the MASP was expected from the entity of the terminal groups and only a slight acceleration of the curing process was observed from 20% of conversion. The addition of a modifier increased the viscosity of the mixture which retarded the reaction because of an immobilizing effect of the reactive species in the mixture. On the contrary, upon increasing the percentage of the modifier, the initiator proportion per epoxy group increased and led to increased reactivity. Both factors influenced the kinetics of the curing process.

## 4. Thermal Properties

The first attempts to enhance the mechanical properties of epoxy thermosets by introducing HBPs achieved significant improvements in tensile strength, flexural strength, and impact strength but at the expense of a decrease in the *T_g_* and thermal stability [69,71,105,106,107]. Theoretically speaking, the modification of epoxy thermosets with HBPs may lead to an increase in the crosslinking density and thus to an increase in the *T_g_*. However, the inherent steric hindrance of the HBP structure often leaves unreacted functional groups in the system. When the miscibility of the HBP into the epoxy matrix is good enough, there is no phase separation and the reduction in the *T_g_* is attributed to a decrease in the crosslinking density. However, if the compatibility between the HBP and the epoxy resin is poor, the system becomes phase-separated upon curing. In many cases, the HBP is compatible with the epoxy at the cure temperature used and it is perfectly dissolved. However, when the homogeneous mixture of epoxy and HBP is cured, the epoxy groups often react faster with the curing agent than the HBP. This results in the formation of hydroxyl groups which decreases the compatibility of epoxy and HBP leading to the formation of a two-phase microstructure [44]. In some cases, the flexible aliphatic structure of the HBP reduces the *T_g_* value despite the crosslinking density achieved [80,108].

DMA analysis is a useful technique to study phase segregation through tan δ vs. temperature plots. As the temperature increases the tan δ goes to a maximum, which is associated with the glass transition temperature *T_g_* and decreases again in the rubbery region. Below the *T_g_*, the damping behavior is low because the polymer chains are frozen, and the deformation is primarily elastic. Above the *T_g_*, the damping behavior is also low because in the rubbery region chains are free to move and their resistance to flow is very low. In the glass transition region, the damping behavior is high because of the micro-Brownian motion of the polymer chains [48]. Neat epoxy formulations show a single peak associated with its *T_g_*, but when some wt.% of HBP is added, if there is phase segregation, two domains are formed: an epoxy-rich domain and an HBP-rich domain. In such cases, another peak is observed at lower temperatures. This β-relaxation is associated with the *T_g_* of an HBP-rich phase. When the percentage of HBP is relatively low (<20%) the *T_g_* of the epoxy-rich phase does not change with HBP content but the *T_g_* of the HBP-rich phase decreases with increasing HBP content. This means that the epoxy resin dissolved in the HBP-rich phase is not sufficiently cured [46]. With high HBP contents, a decrease in the *T_g_* of the epoxy-rich phase can be observed because there is HBP dissolved in the epoxy matrix that prevents the system from being fully cured or plasticized.

The reasons that might cause bad HBP/epoxy interaction are the structure (molecular weight, terminal groups, and polymer structure) and the content of the HBP. In the work of Cicala et al. [106], one can observe all these phenomena. A reduction in the *T_g_* was observed in different epoxy systems by using HBPs with different generation numbers (Boltorn^®^ H30 and H40) and different wt.% content (15% and 30%). SEM micrographs did not show any phase separation and the DMA thermograms did not show any bimodal relaxation curve in formulations with 15% of H30. In this case, the reduction in the *T_g_* can be attributed to a decrease in the crosslinked density in comparison with neat epoxy because of the steric hindrance of the HBP. When using 30% of H30, the system became phase-separated and did so when the system was modified with H40 (with 15 and 30 wt.%). Thus, increasing the content and the molecular weight led to a lower miscibility. When H40 was added, the reduction in the *T_g_* of the epoxy-rich phase was smaller compared to the blends modified with H30 but the β-relaxation increased significantly. The authors stated that H40 was less compatible with the epoxy because fewer polar OH groups, which contributed to the miscibility of the polymer, were exposed on the shell due to the folding-back of the dendritic arms.

Cicala et al. [109] studied the effect of the generation number on the morphological and thermal properties of some epoxy/HBPs thermosets. The term generation is often used to describe the size of HBPs. Each generation represents one repetitive step when building a hyperbranched structure [109], but strictly speaking, generation number can only be applied to dendrimers and not to HBPs, which are synthesized in an only step synthetic procedure. Consequently, if the molar mass increases, maintaining the repetitive unit, the number of reactive chain-ends per molecule increases as well. These authors prepared different samples including three HBPs with different generation numbers and the HBPs were added at 10 wt.% and 20 wt.%. The DMA results showed two peaks in the tan δ curve: one main peak at high temperature associated with the epoxy-rich phase and one less-intense peak in the sub-ambient region associated with a hyperbranched-rich phase. For all formulations containing the hyperbranched modifier, the *T_g_* of the epoxy-rich phase was lower than the *T_g_* of the neat epoxy formulation. This decrease is more notorious when the HBP with the lowest generation number was added because of the higher compatibility with the epoxy network.

In another study, Dhevi et al. [110] synthesized four different hyperbranched polyesters with increasing generation numbers. The curing of the epoxy resin was carried out by reacting the HBP with epoxy resin using hexamethylene diisocyanate, which reacts both with epoxides and alcohols resulting in HBP polyurethane/epoxy-g-interpenetrating polymer networks. The thermal stability of the samples decreased linearly with increasing generations of HBP because of the presence of thermally weak and flexible polyurethane linkages in the modified epoxy samples. The onset temperature lowered from 339 °C in the neat epoxy formulation to 316 °C for the thermoset modified with the HBP of the 4th generation. The HPB-modified samples also exhibited lower glass transition temperatures in comparison to the neat epoxy sample. The authors attributed this trend to the existence of flexible polyurethane linkages which reduced the effective crosslinking density leading to an increase in the free volume for molecular relaxation.

Epoxy resins possess high thermal stability caused by their densely crosslinked network achieved during the curing reaction. However, this thermal stability can be a drawback for the reworkability, for instance, of electronic devices. Thermosetting materials cannot be recycled but they can be degraded under controlled conditions to remove them from the substrate, enabling the recovery of electronic components. Reworkability can be achieved by the introduction of thermally cleavable linkages such as esters of secondary or tertiary alkyl groups in the network structure which can degrade easily upon heating but with a decrease in thermal and mechanical properties [111,112]. A different approach consists of the use of comonomers with ester groups which can be incorporated into the network structure, facilitating the thermal degradation of the resulting materials [113,114]. Based on this approach, our research group reported the use of hyperbranched poly(ester amide)s to enhance the thermal degradability of epoxy thermosets [56,115]. A significant lowering in the thermal stability of the materials studied was observed because of the presence of internal ester groups in the HBP. Other thermomechanical properties were not compromised, and impact strength was even increased.

Although hyperbranched aromatic polyesters also contain a high proportion of ester groups, no decrease in the thermal stability is often observed because aromatic and primary aliphatic esters are not as easily degradable as labile aliphatic tertiary esters [116]. In such cases, chemical reworkability in a basic hydrolytic medium is a good option. Foix et al. [117] and Tomuta et al. [118] evaluated the chemical degradability of HBP modified epoxy thermosets upon saponification in KOH/ethanol under reflux and followed the degradation monitoring of the *T_g_*. In both studies, a progressive decrease in *T_g_* was observed upon saponification time which indicated that the network structure was modified by the incorporation of the HBP containing chemically labile groups.

The modification of epoxy resins with MASPs additives can influence the thermal behavior differently, depending on the chemical structure of the stars. Morell et al. [98,99] synthesized MASPs with different HBP core and poly(ε-caprolactone) arms and they were used as modifiers in the anionic polymerization of DGEBA with 1MI as catalyst. The addition of the MASPs did not modify the processability of the formulation and the *T_g_* was not influenced by the arm length of the MASP, but it decreased slightly on increasing the amount of MASP in the formulation. The presence of ester groups in the arms led to a reduction in the initial degradation temperatures that favored thermal reworkability. When hyperbranched poly(ethyleneimine) was selected as the core in the multiarm star of poly(ε-caprolactone) arms of different lengths, the initial degradation temperature of the final thermoset was slightly reduced on increasing the length of the arms, and the proportion of modifier in the thermoset [102]. In stars with hyperbranched poly(ethyleneimine) core and poly(lactide) arms, the *T_g_* of the thermosets slightly decreased on increasing the amount of the star modifier in the formulation, but there is no change on increasing the number of arms of the MASP. However, the initial degradation temperatures decrease as increasing the amount of the star modifier in the formulation and also on increasing the number of arms [119].

## 5. Thermomechanical Characteristics

Epoxy thermosets are amorphous and highly crosslinked materials. The crosslinking density is a key parameter in determining the mechanical properties of an epoxy resin after cure. The degree of crosslinking and the nature of the bonds in the cured epoxies give them many desirable characteristics, such as good mechanical and thermal properties, that have placed these thermosets as the standard option for a variety of applications e.g., adhesives, coatings, or matrix for composites. However, they have some disadvantages that limit their use in structural applications. Epoxy resins are brittle in general due to their highly crosslinked structure, which confers them low impact strength and poor resistance to crack propagation. The tight structure implies a shrinkage that is undergone during the curing process, which finally leads to the apparition of stresses and defects in the material. In coatings technology, scratch resistance is another issue that needs to be improved. Therefore, new strategies to overcome these drawbacks are needed but without compromising other properties [29].

Toughness refers to the energy that a material can absorb before failure occurs [120]. Thus, toughness implies energy absorption and can be achieved through various deformation mechanisms such as a reduction in the crosslink density, the use of plasticizers that lead to increased plastic deformation, adding toughness modifiers, or incorporating inorganic fillers [26,121]. However, these approaches may seriously affect the modulus and thermal properties of the material and worsen the processability because of an increase in their viscosity. Until now, the most effective toughening mechanism is induced by the addition of a second dispersed phase in form of micro- or nanoparticles that absorb the impact energy and deflect the crack [26]. A combination of cavitation around the particles with shear yielding in the matrix produces a cooperative effect in the energy dissipation [122]. The toughening effect of particles depends on their size, interparticle distance, distribution, particle/matrix interaction, and volume fraction [123].

Toughening by micro/nanostructures can be achieved through chemically induced phase separation methodology (CIPS) from an initial homogeneous blend composed of the resin, curing agent, and modifiers. CIPS is an important method to produce phase-separated morphologies, which generally proceeds from an initially miscible solution to a regular phase-separated morphology during the curing reaction [29]. The reactive functional groups present in the HBP structure also permit the introduction of nonpolar moieties, which reduce the polarity of HBPs and lead to a decrease in their miscibility with epoxies and phase-separated materials [29]. Phase-separated morphologies in blends depend on the kinetics of the curing reaction and the dynamics of the phase separation process. From a thermodynamic point of view, the driving force for the CIPS is the unfavorable entropic contribution to the free energy of the blend resulting from the increasing molecular weight of the epoxy structure during curing [124].

The incorporation of HBPs can improve the toughness of epoxy thermosets without affecting other properties [69]. The improvement in toughness due to HBPs is attributed either to the flexibilizing effect induced by the homogeneous incorporation of the HBP or to the local inhomogeneities created by the formation of phase-separated micro/nanoparticles with good interfacial adhesion between phases. The densely branched core/shell structure of HBPs gives them relatively low viscosity and high molecular weight. Both factors are extremely important in terms of processability. The toughening agent must be compatible with the uncured thermosetting resin to obtain a homogeneous mixture during the initial stage of processing and then become phase-separated only upon curing. Thus, a low viscosity and a relatively high molecular weight toughening agent will facilitate the miscibility of the initial blend and the control of the phase separation process [26]. The solubility of the HBP in the matrix can be controlled by adding polar or non-polar units and then obtaining an initially homogeneous blend which phase separates at a later stage [125]. The mechanical properties of HBPs depend on their bulk structure and on the chemical nature of the shell which controls their compatibility with the surrounding matrix material. The large number of reactive functional groups present in the structure of HBPs enables the grafting og optimal chemical units onto the shell which ensures a high particle/matrix interaction and a good load transfer from the epoxy matrix to the modifier [27].

Generally speaking, the phase-separated toughening effect of HBPs increases to a maximum and then decreases with respect to the particle size domain. The toughening effect in phase-separated morphologies is due to the cavitation of rubbery HBP particles and shear yielding of the matrix. Cavitation absorbs energy and encourages the yielding of the polymer matrix. After the cavitation process, the particles can induce large stress concentrations and extensive shear deformation which is a high energy-absorbing mechanism [126,127]. Both mechanisms can be observed in fracture surfaces due to SEM microscopy. SEM micrographs show more ductile fractures with increasing content of HBP and “sea-island” structures and stress-whitened zones around the crack. Stress whitening is due to the scattering of visible light from the layer of the scattering centers, which in this case are voids. The generation of the voids is due to the cavitations of rubbery HBP particles. The ‘‘sea-island’’ structures reveal that the crack propagation occurs mainly by a mechanism of microvoid coalescence [127]. The stress-whitening phenomenon demonstrates that there exists the crazing of the shear-yield mechanism under impact [128]. These cavities represent the initial position of dispersed particles in the epoxy matrix which were pulled out during the fracture [69]. HBPs have a particular ability to form a property gradient within each phase separated HBP particles with the polymer matrix and thus preventing rigid singularities at particle/matrix interfaces. This means when the HBP dispersed particles have a good interaction with the matrix the ability to transfer stress to the particle increases [26].

Boogh et al. [26] studied the toughening effect of different epoxy-modified hyperbranched polyesters on an epoxy resin cured with tertiary amine as a function of their molecular size, polarity, and epoxy functionality. In the case of epoxy functionalized HBPs, for phase separation to occur, the epoxy groups of the resin and the epoxy groups of the HBP must have different curing kinetics. Polar and reactive hyperbranched molecules were grafted onto the shell and did not phase separate upon curing. In those cases, the toughening effect remained moderate: the stress intensity factor, *K_Ic_*, increased from 0.63 MPa $\sqrt{m}$ for the neat resin to 0.83 MPa $\sqrt{m}$ for the resin system with 25 wt.% of HBP. The hyperbranched modifiers, which were less reactive, induced a phase separation and led to a finely dispersed particulate structure. This led to a 2.5-fold increase in the *K_Ic_* values, from 0.63 MPa $\sqrt{m}$ for the neat resin to 1.54 MPa $\sqrt{m}$ for the modified resin with just a 5% of HBP. This represents a 6-fold increase in the energy release rate *G_Ic_* from 120 J/m^2^ for the neat resin to 720 J/m^2^ for the modified resin. Similar results were obtained in other research studies regarding the toughening effect of epoxy-ended HBPs [27,44,67,68,69,70].

Traditionally, epoxy-ended HBPs showed to be better tougheners for epoxy resins due to their good compatibility compared to hydroxyl-ended HBPs [26]. This was attributed to the fact that epoxy groups are reactive with the epoxy/amine blends while hydroxyl groups do not form covalent bonds at the conventional curing temperatures, which affected the interfacial adhesion [69]. Ratna et al. [48,129] used a commercial hydroxyl-terminated HBP as toughening agent in DGEBA systems cured with diamines. SEM characterizations revealed phase-separated morphologies with HBP particles dispersed in the epoxy matrix. However, the addition of HBP only led to a slight increase in impact strength. The reason was that the HBP did not react with the epoxy matrix and there was no chemical bond between them which affected the interfacial adhesion. Thus, only hydrogen bonds were formed, and the toughening effect remained moderate. Increasing the toughness of DGEBA thermosets with low reactive HBPs may decrease other mechanical properties to different extents which could be attributable to the lower crosslinking density of modified materials [66].

Although the mechanical performance in phase-separated HBP-modified epoxy systems is good in general, sometimes the toughening effect is dependent on the final phase-separated morphology, which is often difficult to control during processing. In addition, phase-separated morphologies can lead to unwanted interfacial effects, which is unacceptable for certain electronic and damping applications, and could also complicate the processing of the thermosets for applications when fibers or other reinforcements have to be added [27,68]. Thus, for those applications, it is necessary to use modifiers that can improve mechanical performance and thermal properties without forming phase-separated morphologies [130].

Several research studies showed that toughness and other mechanical properties of epoxy thermosets could be simultaneously improved by adding compatible HBPs which did not lead to phase separation, such as by using ionic initiators, epoxy/anhydride systems catalyzed by tertiary amines, or by changing the surface structure of the HBP. From 2009, our research group adopted a new curing method to obtain homogeneous morphologies based on epoxy/anhydride systems modified with hydroxyl-terminated HBPs and tertiary amines as initiators [53,56]. Using anhydrides as curing agents, the compatibility between the epoxy matrix and HBPs was improved. Anhydrides can chemically react with both epoxy and hydroxyl groups forming ester linkages. Morell et al. [56,115] used the commercial hyperbranched poly(ester-amide)s Hybrane^®^ S1200 and H1500, as well as another one synthesized by the authors with higher molecular weight, in epoxy/anhydride systems. In general, the addition of HBPs led to an improvement in impact strength (up to 60% with respect to neat epoxy). This increase in toughness was attributed to the flexible structure of the HBPs introduced in the network. It was also observed that the addition of HBPs led to a slight increase in Young’s modulus of the thermosets (up to 10%) and microhardness values (up to 25%).

Zhang’s research group reported the use of different aromatic epoxy-ended HBPs obtained by condensation of trimellitic anhydride and diethylene glycol or butanediol glycol and further glycidation of the free carboxylic groups, which increased the toughness of DGEBA as well as tensile and flexural strength [105,131,132]. Only one *T_g_* was found in DMA curves of the systems modified with second-generation HBPs, indicating a homogeneous structure without any phase separation. The small molecular size of these HBPs helped them to disperse homogeneously in the DGEBA matrix. The epoxy groups of the HBPs and DGEBA reacted with the amine groups of the curing agent. The external crosslinking structure of the HBPs restricted the movement of the internal end-groups of the HBP and could not react with any group. Thus, many non-crosslinked intramolecular cavities absorbed energy, distorted, and formed protonema during impact, indicating high toughness. In addition, the HBP could penetrate the molecular chains of the DGEBA matrix due to its small hydrodynamic volume for increasing interactions between molecular chains during curing and reinforced tensile and flexural strength. Similar results can also be observed in other research studies regarding the use of epoxy-ended HBPs [130,133,134]. Miao et al. [135,136] synthesized an epoxy-ended hyperbranched polyether sulphone as a modifier for DGEBA resin cured with triethylenetetramine, which did not phase-separate. The addition of a medium molecular weight HBP significantly improved toughness, elongation at break, tensile strength as well as *T_g_*. The simultaneous improvements are explained by the higher crosslinking density and increased nanoscale cavities. However, at higher loadings of HBPs, the mechanical performance decreased due to incomplete cure. Similar results were obtained in the study by Zhu et al. [137], in which they also rationalize the results in terms of more flexible crosslinks.

Interesting research papers about the toughening effect of other types of HBPs (apart from epoxy, hydroxyl- or amine-ended HBPs) have been reported in recent years. Fei et al. [90] used a carboxyl terminated HBP (Figure 11) as toughener in epoxy/anhydride systems. The authors obtained an increase in impact strength of 178% with wt.% 5 of HBP in comparison with neat epoxy and an enhancement in tensile strength and elongation at break. These good results were attributed to the synergistic effect of the flexible backbone structure of the HBP and to the increased crosslinked density of modified thermosets since carboxyl groups could participate in the curing reaction. Liu et al. [108,138] synthesized a hyperbranched polysiloxane and used it to modify epoxy/amine and epoxy/phenolic systems. With contents of HBP around 5 and 10 wt.%, impact strength was increased up to 60% in both systems and enhancement of tensile strength, tensile modulus, elongation at break, and flexural modulus were also obtained in comparison with the neat epoxy system. The ductile silicon–oxygen bonds and large amounts of nanoscale cavities also provided the HBP-modified epoxy thermosets with improved toughness. The large amount of rigid benzene rings in the HBP structure and the higher crosslinking density contributed to the enhancement of mechanical properties.

Lu et al. [139] synthesized a series of thiol-ended HBPs with different amounts of thiol groups as curing agents to crosslink DGEBA through a thiol-epoxy click reaction (Figure 16). A general mechanical performance improvement was obtained on the cured materials thanks to the synergetic effect of hydroxyl groups present in the structure, intramolecular cavities, crosslinking density, dispersion of the HBP in the matrix, and topological structure of the crosslinked network.

Park and Choe [140] synthesized a hyperbranched poly(methylacrylate-diethanolamine) (poly(MA-DEA)) and a poly(methylacrylate-ethanolamine) (poly(MA-EA)) (Figure 17) and used them to modify a DGEBA resin cured with Jeffamine^®^ D400. In general, a steady decrease in *T_g_* was observed upon increasing the content of the HBPs because of a reduction in the crosslinking density and network stiffness. Tensile strength showed a maximum with 3 wt.% and 5 wt.% with poly(MA-DEA) and (poly(MA-EA), respectively, and elongation at break increased as increasing the content of HBPs. In the case of impact strength, when the HBP content was higher than 10 wt%, the steric hindrance effect of the HBP inhibited hydrogen bonding and led to a decrease in impact strength.

Several factors affect the performance of HBPs as toughening agents, such as the molecular weight or generation number [109,110,115]. Generally speaking, the addition of HBPs to epoxy thermosets improves tensile, flexural, impact strength, and fracture toughness. These properties are increased by increasing the proportion of HBP and its molecular weight until reaching a maximum. However, this maximum also depends on its chemical structure and cannot be taken as a general trend [115]. 

Morell et al. [115] compared the effect of the molecular weight of two HBPs with cycloaliphatic rings in their structures on an epoxy/anhydride system catalyzed by a tertiary amine. The first HBP was the commercial Hybrane^®^ H1500 (Figure 18) with 1500 g/mol and the second was an HBP with a molecular weight of 30,000 g/mol that they synthesized. From the calorimetric data, it can be observed that the molecular weight barely affected the curing kinetics of the system. If one compares the enthalpy reaction and the peak temperature of formulations with wt.% 5 and wt.% 10 of both HBPs, no significant differences were observed.

Another important issue regarding the mechanical properties of thermosets is shrinkage. Shrinkage is the reduction in volume brought about by an increase in density throughout the curing process of a thermoset. Chemical shrinkage occurs because of the formation of new covalent bonds during the curing process. Usually, the new bonds formed are shorter in the polymer than the distances between monomers. The curing process of epoxy resins is a critical point in the field of coatings since shrinkage can lead to the appearance of internal stresses and defects in the materials, such as microvoids, microcracks, or warping [29]. The global shrinkage during curing can be calculated from the density values of the material before and after curing. Fernández-Francos et al. [50] reported a reduction in contraction during curing (and even an expansion) when Boltorn^®^ H30 was used to modify a DGEBA system cured with Yb(OTf)_3_ to chemically incorporate the HBP into the epoxy matrix. This result was attributed to the higher compactness of Boltorn^®^ H30 before curing produced by the large number of intermolecular H-bond and intramolecular H-bond interactions which decreased when the terminal hydroxyl groups reacted with the epoxy groups. It was also reported a reduction in shrinkage after gelation [53,56], which is the true responsible for the apparition of stresses because the material loses its mobility, and therefore the shrinkage produces tensile, compressive, and shear forces within the resin.

The existence of a great number of terminal groups in the HBP structure allows the compatibility of the HBP to be tailored, and the epoxy matrix which, as said before, would affect phase separation and mechanical performance in an extensive way. Several HBPs were modified in different proportions with 10-undecenoyl groups, which allowed phase-separation or homogeneous materials to be achieved [86]. Hyperbranched polyethyleneimine modified with 78 and 91% of these long chains allowed the reaching of 100% improvement in impact strength with only a slight reduction in the *T_g_* and thermal stability. 10-undecenoyl modified hyperbranched poly(glycidol) was also added to a cycloaliphatic epoxy resin and cationically cured with Yb(OTf)_3_ [83]. The modification degree of the OH groups in the HBP was 45 and 80% and the impact resistance of the thermosets reached a 200% of improvement because of the formation of soft microparticles well distributed in the matrix with a certain degree of covalent bonding that improves compatibility. This enhancement was achieved without sacrificing thermomechanical properties or processability. Better results in impact resistance were achieved when Boltorn^®^ H30 was modified with several proportions of 10-undecenoyl chloride [141]. Four-fold impact resistance of DGEBA/anhydride thermosets was reached by adding 10% of HBP with a 76% of modification degree without affecting thermal stability, thermomechanical characteristics, or processability. The tailoring of the chemical structures of Boltorn^®^ H30 polyesters allowed a regular microphase separation to be reached with excellent interfacial interaction, which leads to a cavitation effect on the fracture mechanism.

Although MASPs have been less used to improve the mechanical characteristics of epoxy thermosets, there are some studies in this field. Ren et. al. [142] synthesized 3, 4, and 6-arm stars with glycidyl methacrylate arms by ATRP. The epoxy groups in the arms and the epoxy resin could react with 4,4′-diamino diphenyl sulphone and the materials obtained presented higher impact resistance than their linear analogous. The 4-arm star led to the highest toughness effect.

In our research team, a broad exploration has been carried out on the use of MASPs as toughness modifiers. Higher impact resistance, in comparison to the linear analogous, was reached when MASP with poly(glycidol) core and poly(ε-caprolactone) arms were added to DGEBA cured by Yb(OTf)_3_ [98]. The impact strength improvement of the thermosets was increased three times when hydroxyl groups at the end of the poly(ε-caprolactone) MASPs were end-capped with acetyl groups, without an appreciable decrease in *T_g_*, but with a slight decrease in microhardness and elastic modulus [143]. Amphiphilic MASPs with poly(ethylene glycol) and poly(ε-caprolactone) arms led us to double the impact resistance of epoxy/anhydride matrices, showing a phase-separated morphology which led to an increase in the energy for fracture [17].

## 6. Rheological Properties

Some rheological parameters, such as viscosity, shear response, and pot life, have enormous importance in the processability, flowability, and stability of the epoxy system. One of the main advantages of HBPs is that they show low viscosity in comparison to their lineal counterparts with the same molecular weight per equivalent functional group. The variation in the viscosity of the blends depends on the difference in viscosity between the epoxy matrix and the HBP and the structure of the HBP. In some cases, even a small content of HBP may significantly increase the viscosity of the system [52,95,144]. However, depending on the aliphatic nature of the HBP branches, a significant decrease in viscosity can be obtained due to low molecular entanglements [108,145,146]. In some cases, aside from the change in viscosity, the addition of HBP causes the blends to show a non-Newtonian behavior [147]. HBPs show shear-thinning behavior, that is, their viscosity decreases with shear. This behavior is thought to be caused by the hydrogen bonds between the HBPs branches which are destroyed under shear forces [108,148]. 

Although gelation is not a rheological phenomenon itself, rheology provides useful tools to determine the gel time. Gelation corresponds to the formation of a giant macromolecular structure that percolates the reaction medium. The gel fraction extends and increases crosslinking of the system as the curing advances leading to a fully three-dimensional network. Macroscopically, a change from liquid to solid mass can be observed; the material ceases to flow and starts to build up mechanical properties [28]. The gelation time can be determined by the performance of time sweep multiwave oscillation experiments either isothermally or with a temperature ramp. The gelation time can be determined as the tan δ crossover of the different harmonics [149]. In addition, the conversion at the gelation can be determined by stopping the rheological experiment at gelation and performing a DSC scan of the gelled sample, and comparing the enthalpy released with the curing enthalpy of the initial formulation. The effect of HBPs on the gelation process is not clear and depends on the viscosity and reactive functional groups of the HBP and the chemistry of the curing process. Because of their high number of reactive functional groups, the addition of HBPs may cause a decrease in gelation time but an increase in conversion at the gelation [52,55,86,96]. The decrease in gelation time and the possible increase in the viscosity of the system can be an issue regarding the processability of epoxy/HBP blends. However, the increase in the conversion in the gelation should have a positive effect in terms of internal stresses because fewer tensions are generated within the matrix, assuming that curing shrinkage is proportional to conversion [150].

Regarding the viscosity of formulations, the addition of MASPs as modifiers in epoxy resins leads to smaller increases than their linear analogs. Ren et al. [142] could observe that the viscosity of the formulations decreased on increasing the number of arms in well-defined stars and that the effect was lower than in the case of adding an analogous linear polymer. This result was attributed to the lower inter and intra-macromolecular entanglement of the MASP in comparison to the linear counterpart. In our research work on this topic, we could prove by rheology that multiarm copolymers with a higher number of arms and shorter length arms led to a decrease in the complex viscosity which indicates better processability in the melt dynamic state [102,151].

## 7. Fire Retardancy

Besides the inherent brittleness of epoxy resins, another disadvantage of these materials which could limit their application is that epoxy resins are more combustible than similar thermosets because of their reduced tendency to carbonize [152]. The fire resistance of epoxy resins can be improved by incorporating bromine-containing additives. However, they release hydrogen bromide, dibenzo-p-dioxin, and dibenzo-furan during combustion which can lead to corrosion and toxicity problems. Recent developments in the chemistry of halogen-free flame-retardant polymers involve reactive monomers that are inherently flame retarding, such as those containing P, Si, B, or N [153,154,155,156]. The incorporation of flame retardants usually led to a reduction in physical properties due to the poor dispersibility and the high contents needed [157,158,159]. Thus, the motivation of researchers has been to develop flame-retardant modifiers to overcome flammability but without damaging other properties, such as *T_g_* [160,161]. In this sense, several research papers about phosphorus-containing HBPs can be found in the literature [162,163,164,165]. In this section, some of the most recent studies of phosphorus-containing hyperbranched flame retardants are reviewed.

Chao et al. [166,167] synthesized a hyperbranched poly(aminomethylphosphine oxide-amine) (HPAPOA) and a hyperbranched poly(urethane-phosphine oxide) (HPUPO) and improved flame retardancy and toughness when they were incorporated into epoxy systems. In the case of HPAPOA (Figure 19) [166], when 3 wt.% was added, the epoxy thermoset passed the UL-94 V-0 rating with a LOI value of 30.7% due to the blowing-out effect. In addition, *T_g_* was raised due to the increased crosslinking density and impact strength showed a slight increase due to the in situ energy dissipation of intra-molecular cavities and inter-molecular free volume between HBP molecules. In the case of HPUPO [167], when the content of HPUPO content increased, the LOI value and UL-94 rating of modified epoxy thermosets progressively improved, and with 4 wt.% HPUPO, EP4 reached a V-0 rating with a LOI value of 30.5%

Schartel’s group worked intensively on phosphorus based flame retardants, including the utilization of HBPs [168,169,170,171,172]. Hyperbranched phosphorous-containing flame retardants were found to be more efficient than an oligomeric phosphate flame retardant because of their higher phosphorus content and higher efficiency per phosphorus atom. Battig et al. [171] compared the efficacy of previously reported hyperbranched flame retardants in two distinct epoxy resins, one aliphatic and one aromatic, with the aim of deepening the understanding of the flame retardant-matrix co-dependency on effective flame retardancy. The authors stated the importance of the decomposition temperature (*T_dec_*) overlap. When this *T_dec_* overlap between flame retardant and matrix was higher, the chemical interaction was greater and led to effective flame retardancy in the form of higher char yield, reduced peak of heat release rate, and lower total heat release. Hyperbranched flame retardants offer the possibility of tuning the *T_dec_* via molecular weight and thus the overlap temperature range can be adjusted. In another study, Battig et al. [172] compared the flame retardant potential of different HBPs with different contents of P-O and P-N bonds in terms of flame retardancy mechanisms, mode of action, and efficacy in epoxy resins. The residue yield after pyrolysis increased linearly with the N-content of hyperbranched flame retardants and those with a higher N-content were thermally more stable than those with a higher O-content. 

Hu et al. [173] synthesized a hyperbranched phosphorus/nitrogen-containing flame retardant (Figure 20) via the transesterification reaction of dimethyl methylphosphonate and tris (2-hydroxyethyl) isocyanurate and it was used to obtain epoxy thermosets with high fire retardancy and transparency. The sample with 4 wt.% of hyperbranched flame retardant achieved a V-0 rating in the UL-94 test and showed a high LOI value of 34.5%. The addition of the hyperbranched flame retardant shifted the temperature at a maximum decomposition rate to a higher value and enhanced the char yield at 700 °C. In addition, modified thermosets kept high values of transmittance around 85% in the visible region.

Döring’s research groups have published several research papers on flame retardancy of epoxy-based composites and thermosets, including MASPs phosphorous-containing flame retardants [164,174]. In the work by Müller et al. [174], the authors synthesized six different star-shaped phosphorous-containing flame retardants and used them to modify RTM (resin transfer molding) resins. The relatively low viscosity of MASPs makes them a good modifier since viscosity is a key parameter in RMT applications. The authors combined six phosphorous-containing MASPs with three RTM systems and, in general, they found good results, according to UL 94-V classification.

Finally, to summarize all the content of this review, Table 1 enlists the most relevant works regarding the enhancing of epoxy thermosets with HBPs and MASPs.

## 8. Conclusions

HBPs and MASPs have been demonstrated to be excellent modifiers for increasing the toughness of epoxy thermosets without affecting other properties. This review has covered the effect of hyperbranched polymers and multiarm star polymers on the modification of epoxy thermosets regarding the effect on the curing process, thermal, mechanical, and rheological properties, and fire retardancy through the most relevant studies on the topic. They can be relatively easily obtained through a one-step polymerization reaction and their structure adjusted to obtain the desired properties of final materials.

There is not a general trend regarding the effect of HBPs and MASPs on the physical properties of epoxy-based thermosets because they depend on the type and content of HBPs or MASPs. For instance, the curing process is affected by several factors. The reactive functional groups of HBPs or MASPs can enhance the curing rate of the system but, at the same time, topological restrictions, and the increase in the viscosity of the formulation can worsen the curing rate. Thermal properties also depend on the type and content of HBP or MASP. The *T_g_* can increase due to an increase in the crosslinking density but, if the compatibility between the epoxy and the modifier is poor, HBP or MASP can phase separate and thus the *T_g_* of the epoxy-rich domain does not change with HBP or MASP content. However, the *T_g_* can decrease with HBP or MASP content if they have an aliphatic flexible structure. The incorporation of HBPs or MASP can improve the toughness of epoxy thermosets due to the flexibilizing effect induced by the homogeneous incorporation of them, or due to the local inhomogeneities created by the formation of an HBP/MASP rich separated phase. Due to their tunable structure, HBPs and MASPs can be synthesized to introduce properties at demand, such as increased chemical reworkability by introducing ester groups, or increased fire retardancy due to phosphorous-based HBPs or MASPs.

## Data Availability

Not applicable.

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
