# Peer review of "Enhancement of Epoxy Thermosets with Hyperbranched and Multiarm Star Polymers: A Review"

_polymers, 2022, doi:10.3390/polym14112228_

Round 1
Reviewer 1 Report
This review reports the behaviour of epoxy thermostes with hyperbranched and multiarm star polymers. The paper is clearly written. The paper is ready for publication in Polymers
Author Response
Dear Reviewer;
Thank you very much for your time and your kind words.
Best regards,
David Santiago
Reviewer 2 Report
Dear editor, dear authors,
This review presented by Santiago & Serra is a nice piece of literature that may deserve to be published already in the present version.
The only suggestion I give the authors is the possibility to introduce, just before the conclusions, one table which summarize the finding of every research, by highlighting the main finding and the key-message of the article. This would increase dramatically the future citations of this work and will facilitate the reader to quickly find the part which interest the most.
I have found only few typos, all at the beginning of the article:
- Line 26: acronym should be HBP
- Line 30: link with figure 1 – technical problem
- Line 43: comma missing
The software detected only 7% plagiarism (self) which confirms the originality of this work.
The more similar work was this:
Acebo, C., Picardi, A., Fernández-Francos, X., De la Flor, S., Ramis, X., & Serra, À. (2014). Effect of hydroxyl ended and end-capped multiarm star polymers on the curing process and mechanical characteristics of epoxy/anhydride thermosets. Progress in organic coatings, 77(8), 1288-1298.
Which the authors did not cite (indeed they cited already several works of their own group, but this may also add further information and I suggest to add it to the reference list).
I recommend the editor considering this paper to be ticked as particularly interesting.
Author Response
Dear reviewer;
Thank you very much for your time.
We introduced a summary table before the conclusions, as you suggested. We also corrected the typological errors at the beggining of the article.
Regarding the work:
Acebo, C., Picardi, A., Fernández-Francos, X., De la Flor, S., Ramis, X., & Serra, À. (2014). Effect of hydroxyl ended and end-capped multiarm star polymers on the curing process and mechanical characteristics of epoxy/anhydride thermosets. Progress in organic coatings, 77(8), 1288-1298.
It appears in the reference list: 143.
I would like to thank you again for your time and your kind words.
Best regards;
David Santiago